# Knowledge sharing behaviour among head nurses in online health communities: The moderating role of knowledge self-efficacy

**Salah Shehab**[1]*, **Mohammad Al-Bsheish**[2,3]*, **Ahmed Meri**[4], **Mohammed Dauwed**[5], **Badr K. Aldhmadi**[6], **Haitham Mohsin Kareem**[7], **Adi Alsyouf**[8], **Khalid Al-Mugheed**[9], **Mu'taman Jarrar**[10,11]*

**1** College of Graduate Studies, Universiti Tenaga Nasional, Kajang, Selangor, Malaysia, **2** Health Management Department, Batterjee Medical College, Jeddah, Saudi Arabia, **3** Al-Nadeem Governmental Hospital, Ministry of Health, Amman, Jordan, **4** Department of Medical Instrumentation Techniques Engineering, Al-Hussain University College, Karbala, Iraq, **5** Department of Medical Instrumentation Techniques Engineering, Dijlah University College, Baghdad, Iraq, **6** Department of Health Management, College of Public Health and Health Informatics, University of Ha'il, Ha'il, Saudi Arabia, **7** Department of Accounting, Southern Technical University, Basrah, Iraq, **8** Department of Managing Health Services and Hospitals, Faculty of Business Rabigh, College of Business (COB), King Abdulaziz University, Jeddah, Saudi Arabia, **9** Nursing Department, Health Faculty, Riyadh Elm University, Riyadh, Saudi Arabia, **10** Vice Deanship for Development and Community Partnership, Imam Abdulrahman Bin Faisal University, Dammam, Eastern Province, Saudi Arabia, **11** Medical Education Department, King Fahad Hospital of the University, Al-Khobar, Saudi Arabia

* dr.salah-shehab@email.com (SS); mohammed.ghandour@bmc.edu.sa (MAB); mkjarrar@iau.edu.sa (MJ)

**Data Availability Statement:** All relevant data are within the article and its Supporting Information files.

## Abstract

### Background

Head nurses are vital in understanding and encouraging knowledge sharing among their followers. However, few empirical studies have highlighted their contribution to knowledge-sharing behaviour in Online Health Communities (OHCs). In addition, scant literature has examined the moderating role of knowledge self-efficacy in this regard.

### Purposes

This study examines the moderating role of self-efficacy between the association of four selected individual factors of head nurses (i.e., Trust, Reciprocity, Reputation, and Ability to Share) and their knowledge-sharing behaviour in OHCs in Jordan.

### Method

The data were obtained by using a self-reported survey from 283 head nurses in 22 private hospitals in Jordan. A moderation regression analysis using a structural equation modelling approach (i.e. Smart PLS-SEM, Version 3) was utilised to evaluate the study's measurement and structural model.

**Funding:** The authors received no specific funding for this work.

**Competing interests:** The authors have declared that no competing interests exist.

## Results

Knowledge self-efficacy moderates the relationship between the three individual factors (i.e., Trust, Reciprocity, and Reputation) and knowledge-sharing behaviours. However, self-efficacy did not moderate the relationship between the ability to share and knowledge-sharing behaviours.

## Implications

This study contributes to understanding the moderating role of knowledge self-efficacy among head nurses in online healthcare communities. Moreover, this study provides guidelines for head nurses to become active members in knowledge sharing in OHCs. The findings of this study offer a basis for further research on knowledge sharing in the healthcare sector.

## 1. Introduction

Knowledge-sharing behaviour is becoming increasingly indispensable in today's business environment [1, 2]. Knowledge sharing is an essential resource for effectively implementing essential business functions, and like other industries, healthcare organisations are beginning to use knowledge sharing as a new practice. Knowledge sharing is a conveyance behaviour wherein individuals disperse their knowledge, experiences, and skills to others [3]. Effective knowledge sharing is vital in healthcare organisations because it significantly enhances the quality of care and patient safety [4]. Healthcare workers can use knowledge sharing for their patients, making it easier to share information about their diagnoses and treatments [5]. Thus, knowledge sharing is a strong element for improvements and further development within the healthcare sector [6, 7].

The existence of Online Communities (OC) can facilitate knowledge sharing [8, 9]. In contrast to traditional knowledge sharing in real-world communities, members of OC are distributed across geographic locations. Difficulties related to face-to-face knowledge exchanges among OC members may weaken the bond among OC. Therefore, scholars have investigated knowledge-sharing behaviours in various online communities [10–12]. Online Health Communities (OHCs) are one kind of an OC, where maintaining health information is a public concern. OHCs through social media and other web-based forums, facilitate their members to participate in health topics, even those with sensitive considerations such as pregnancy, menstruation, and sexuality [13, 14].

OHCs have recently received substantial attention from health practitioners due to several considerations. Everyday users tend to be well-educated on disease causes, treatment advice and preventive actions by simply inputting personal health information into OHCs [15]. Individuals go as far as to opt for self-diagnosing through OHCs rather than the traditional way by physically visiting hospitals [16]. Besides, OHCs grew impressively after observable internet technology advancement and emerged as a powerful medium among healthcare providers to be active members in OHCs [17].

Participation in OHCs by healthcare workers can share their experiences, information, and feelings with each other and offer help and support [14]. One benefit of OHCs includes 24/7 access to information and assistance from individuals without any restrictions imposed by geographic location. The relatively free and less risk-oriented nature suggests that several opinions are always better for making decisions regarding health and medical concerns [18–20].

Previous studies show that OHCs are positively associated with a user's treatment options, health outlook, and outcomes [21]. Participants who share knowledge within an OHC view the contribution as a perceived benefit as they may find happiness in enhancing their knowledge or social value in educating others [21, 22]. Other perceived benefits may include financial incentives from the communities (such as fees or donations), the joy of interacting with other community members, and/or the increased reputation within the community due to their contributions [23].

The process of knowledge sharing is less effective within an organisation without the involvement and engagement of the human element [24]. Several studies have identified the role of individual factors in knowledge-sharing behaviours [25–27]. For instance, Shehab et al. [28] reviewed 31 studies that investigated predictors of knowledge-sharing in different contexts; they found that the roles of individual factors are dominant. However, knowledge-sharing behaviours studies in Jordanian hospitals are scarce. As Alhalhouli [29] reported, "Variables that enhance or dissuade knowledge sharing behaviours in the Jordanian hospitals have not been poorly recognised." Al-Dalaien et al. [30] established a conceptual model of motivational factors of knowledge transfer in Jordanian hospitals. Aldohyan's et al. [31] study in the Saudi context emphasised that "Hospitals should always refer to efficient knowledge sharing and educational strategies that render beneficial outcomes to patients, healthcare workers, and the public community". However, lack of studies exploring the role of knowledge self-efficacy between the associations of individual factors (trust, reciprocity, reputation, and ability to share) with knowledge-sharing behaviours.

Accordingly, the current study extends the previous literature and fills the gap by examining four individual factors (trust, reciprocity, reputation, and ability to share) with knowledge-sharing behaviours in OHCs. Additionally, this study examines the moderating effect of knowledge self-efficacy as it can change the strength of the direct effect between the above-mentioned individual factors and the knowledge-sharing behaviours of head nurses in Jordan.

This article is organised as follows. The first section discusses the study theories, hypotheses, and research model. This is followed by a section that presents the research methodology and analyses the results. Last, the implications and conclusions have been provided as well.

## 2. Literature review

### 2.1 Underpinning theory

Social Cognitive Theory (SCT) postulates that the mutually triangular interaction of individual factors like individual cognition, social factors such as social group (Online Community), environmental factors, and individual expectations and beliefs shape human behaviours [32, 33]. SCT primarily focuses on self-efficacy, considered as useful prescriptive and practical concepts formulated in modern psychology" [34]. Other authors also provided their opinions on self-efficacy. For example, Lent [35] states that self-efficacy refers to "people's judgment of their abilities to organise and implement courses of action required to achieve certain types of performance". The study moderator (i.e. self-efficacy) lays the foundation for personal achievement, personal well-being, and human motivation, human performance; Bandura [36] assumed that people's level of motivation, emotional states, and actions depend more on what they believe than on what is objectively true. Self-efficacy reflects people's beliefs about their competence or effectiveness in carrying out tasks and tends to be more self-confident [37].

Previous literature has also empirically confirmed this concept [38–40]. SCT suggests that individual motivation and action are apparent bounded, and an individual is more or less likely to undertake a specified behaviour [41]. Thus, the study model used social cognitive theory.

## 2.2 Hypotheses building

Over the last few decades, studies have emphasised the importance of individual self-efficacy and expectation in predicting individual health behaviours [42]. Self-efficacy refers to people's judgment of their capabilities to organise and execute courses of action required to attain designated types of performance [35, 43]. It is "one of the most theoretically, heuristically and practically useful concepts formulated in modern psychology" [34]. Prior research has demonstrated that self-efficacy lays the foundation for personal achievements, personal well-being, and human motivation. Bandura [36] explained, "People's level of motivation, affective states, and actions are based more on what they believe than on what is objectively true".

Knowledge self-efficacy is important in influencing the process of knowledge sharing and the influencing factors that contribute to knowledge sharing among online communities. For example, Hsieh et al.'s study showed that knowledge self-efficacy could moderate the relationships between reputation and pleasure in helping others share knowledge [44]. Thus, the conclusion can be reached that self-efficacy strongly influences an individual's behaviour [45]. Aligned with this, the current study investigates the moderating effect of self-efficacy on the relationship between individual factors and knowledge-sharing behaviour. The assumption is that when the level of knowledge and self-efficacy is high, head nurses are very confident in their ability to provide valuable knowledge.

Knowledge sharing in online communities has been given less attention to the relationship between knowledge self-efficacy and knowledge-sharing behaviour [21]. This may be an issue in knowledge sharing because complexity and knowledge barriers to exchanging knowledge among online communities may be seen as knowledge efficacy deficits [46, 47].

Knowledge self-efficacy suggests that people who think their knowledge is valuable would be more likely to share greater knowledge [48]. It is described as a function of self-beliefs with which individuals accomplish a particular work [49], and knowledge self-efficacy can lead to greater productivity and performance. Knowledge self-efficacy is a type of self-assessment affecting decisions on how an individual will behave and be motivated under tasks and the level of effort asserted in the face of challenges.

Past researchers have linked knowledge self-efficacy to motivation and behaviour [49, 50]. Those with higher levels of self-efficacy tend to perform better than those with lower levels [51]. Recently, researchers have concentrated on knowledge self-efficacy. This has been implemented in knowledge management to validate the effect of self-assessment, self-confidence, and motivation of individuals for knowledge sharing. Self-efficacy is highlighted as individual expectations of positive outcomes of behaviour since, if individuals doubt the capability to complete the behaviour successfully, pursuing an action would be perceived as worthless. According to Wasko and Faraj [52], an individual with high knowledge self-efficacy may feel happy answering questions easily, specifically questions from beginners. Consequently, such a person may develop a more positive behaviour towards sharing knowledge [53–55]. Additionally, their ethical commitment should strongly influence knowledge-sharing behaviour in online healthcare communities.

The current study anticipates that the influence of individual factors of trust, reciprocity, reputation, and ability to engage in knowledge-sharing behaviour will become stronger as head nurses gain more knowledge and self-efficacy in online healthcare communities. (See Fig 1).

Accordingly, the following hypotheses are posited:

H1: Knowledge self-efficacy moderates the relationship between trust and knowledge-sharing behaviour.

H2: Knowledge self-efficacy moderates the relationship between reciprocity and knowledge-sharing behaviour.

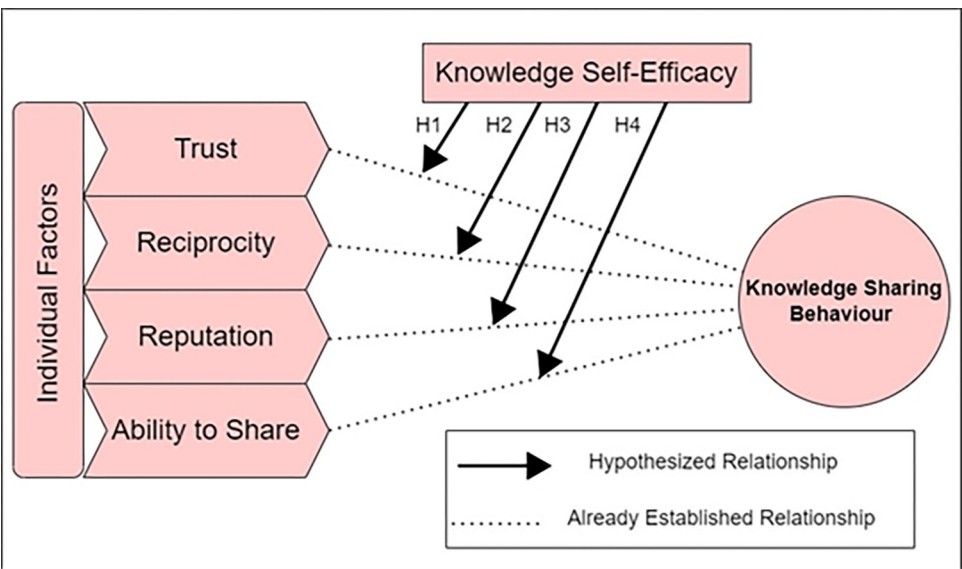

**Fig 1. Research model.**

H3: Knowledge self-efficacy moderates the relationship between reputation and knowledge-sharing behaviour.

H4: Knowledge self-efficacy moderates the relationship between the ability to share and knowledge-sharing behaviour.

## 3. Methodology

### 3.1 Design, sampling, and settings

A quantitative cross-sectional study was conducted using self-reported booklet surveys targeting individual head nurses. The population of this study was private hospitals' head nurses in Jordan (total private hospitals = 68). Head nurses were targeted in this study (Total number of head nurses = 510) to serve the study purposes as they are considered health leaders. Public and even followers' nurses prefer contact with head nurses due to their managerial rank as first-line management; thus, they are knowledgeable, closer, and often participate in online communities. The researchers purposively selected the private hospitals in the capital (i.e. Amman) because these institutions have competitive advantages, technological capabilities, the highest capacity, diversity in terms of speciality, supportive research cultures, and the highest number of hospitals located in Amman. The research team sent the request to all private hospitals in Amman (n = 32 hospitals). Ten hospitals rejected participation in this study. Approval was received from 22 private hospitals with a total of 322 head nurses (study population.

The sample size was calculated based on the G*Power software package, effect size ($f^2$ = 0.15); a significant level ($\alpha$ = 0.05), and power 1-$\beta$ = 0.95, which calculated a minimum sample size of 74 was required with five independent variables, including the moderator. Beside G*Power software package, Krejcie and Morgan, (1970) sample size formula was used to get minimum number of respondents to be surveyed, which is 210" [56]. Therefore, this study reached data from 283 respondents, which was satisfactory.

### 3.2 Ethical considerations and data collection procedures

Ethical approval number UNITEN/COGS 23/2/1/PM20604 was attained from the *College of Graduate Studies*, *Universiti Tenaga Nasional*, Malaysia, on 25 April 2018. The 22 private hospitals approved their employees' voluntary participation in the study and encouraged their head nurses to participate. The first page of the booklet survey was a cover page that provided the study purposes, the necessary definitions, and the approval sign to conduct the study. Written consent was obtained from all the respondents after they were informed regarding their right to withdraw from participation at any time, that data would be only for academic purposes, and that their responses would be confidential. Booklet surveys were distributed personally to all head nurses in the hospitals ($n$ = 22) to be completed by their staff ($n$ = 322). The consent form was gathered from participants; accordingly, the data collection process started in May 2018 and lasted until October 2018. Of the 322 surveys distributed, the total usable surveys received were 283, with an effective response rate of 84%.

### 3.3 Measures

The study questionnaire was revised several times before starting the collection data process (i.e. content validity). The last version of the questionnaire includes two parts; the first asked demographic questions such as gender, age, education, internet usage, and experience. The second part included six scales (i.e., trust, reciprocity, reputation, ability to share, knowledge self-efficacy, knowledge sharing behaviour) and used a 5-point Likert ranging from 1 = strongly disagree to 5 = strongly agree. The first scale was *trust*, defined as employees' belief in good intent, competence, and reliability concerning contributing and reusing knowledge. The four items for the trust scale were adapted from previous literature [57, 58]. The Cronbach's alpha of the items was 0.912. The second scale contained three items adapted from the *reciprocity* scale that Zhang et al. [21] modified and referred to a belief that current sharing behaviour would cause future requests for knowledge to be easily satisfied by others [21]. The Cronbach's alpha of the items was 0.921. The third scale was *reputation*, which refers to a perception of improved reputation and image due to sharing knowledge in the online community. Four items were adapted from Kankanhalli et al.'s [54] study. The Cronbach's alpha of the items was 0.931. The fourth scale was the *ability to share*, which refers to the capabilities of conceiving and sharing meaning in different situations. The scale was adapted from Radaelli et al. [59]. The Cronbach's alpha of the items was 0.86. The fifth scale was *self-efficacy*, which means the degree of confidence in one's ability to provide valuable knowledge to others. The four items were adapted from Bock and Kim [60] and Lu et al. [61]. The Cronbach's alpha of the items was 0.902. The last scale was *knowledge sharing behaviour*, which refers to a process of knowledge exchange between individuals who disperse their obtained knowledge, experiences, and skills to others and groups [21]. Five items were used to measure knowledge-sharing behaviour from Bock and Kim [60] and Lu et al. [61] studies. The Cronbach's alpha of the items was a = 0.85. The survey was written in the English language. S1 Appendix presents a list of items for each of the measures.

### 3.4 Data analysis techniques

Descriptive statistics and moderation regression analysis using SPSS and structural equation modelling approach (i.e. Smart PLS-SEM, Version 3) were the key statistics in this study. This study used Smart PLS3 to test the hypotheses posited. Smart PLS3 uses a bootstrapping technique to estimate path coefficients and standard errors [62]. The moderation impact of self-efficacy was evaluated considering a 5000-bootstrap sample, a 95% confidence interval (CI)

and a significance level of 0.05. Before running Smart PLS3, descriptive results were performed using SPSS Version 18.0 (SPSS Inc., Chicago, IL, USA).

## 4. Results

### 4.1 Demographics characteristics

The descriptive and frequency analysis output of SPSS 18.0 showed the 283 nurses' demographic characteristics. The majority were female head nurses (52.7%), 25–30 years old (30.4%), with Bachelor's degrees (71.4%) and more than 10 years of experience (71%). Furthermore, the daily Internet usage among the head nurses was 1–3 hours (52.7%). (See Table 1).

### 4.2 Validity and reliability

Smart PLS 3.0 measurement or outer models detect if the collected data are valid and reliable. In this study, Convergent validity was tested using Cronbach's alpha (α), Composite reliability (CR), and Average Variance Extracted (AVE) and achieved an acceptable value. Cronbach's alpha, Composite reliability, and Average Variance Extracted were more than .70, .70, and .50; respectively, of the study variables (trust (α = .89, CR = .92 and AVR = .75), reciprocity (α = .89, CR = .93 and AVR = .82), reputation (α = .90, CR = .93 and AVR = .77), and ability to share (α = .92, CR = .94 and AVR = .81), Knowledge Self-Efficacy (α = .93, CR = .95 and AVR = .82), and Knowledge Sharing Behaviour (α = .78, CR = .86 and AVR = .61). Fornell-Larcker criterion and Heterotrait-Monotrait (HTMT) were examined by Smart PLS 3.0 measurement model and indicated to valid data [63]. (See Table 2).

Discriminant validity could also be examined by assessing items' cross-loading [63]. To achieve an acceptable level of cross loading, the indicators' (items) loading of the constructs should be higher than the loading on another construct, which was achieved as Table 3 shows.

**Table 1. Demographic characteristics.**

| Characteristic | Profile | N | % |
|---|---|---|---|
| Gender | Male | 134 | 47.3 |
| | Female | 149 | 52.7 |
| Age | 25–30 years | 86 | 30.4 |
| | 31–35 years | 73 | 25.8 |
| | 36–40 years | 58 | 20.5 |
| | > 40 years | 66 | 23.3 |
| Education | Bachelor's | 202 | 71.4 |
| | High diploma | 45 | 15.9 |
| | Masters | 33 | 11.7 |
| | PhD | 3 | 1.1 |
| Internet usage | < 1hour | 27 | 9.5 |
| | 1–3 hours | 149 | 52.7 |
| | 4–6 hours | 66 | 23.3 |
| | >6 hours | 41 | 14.5 |
| Experience | <5 years | 17 | 6 |
| | 5–10 years | 65 | 23 |
| | >10 years | 201 | 71 |
| Total | | 283 | |

**Table 2. Convergent and discriminant validity.**

| Constr. | Convergent validity | | | *Discriminant validity (Fornell-Larcker criterion) | | | | | | **Discriminant validity (HTMT Ratio) | | | | | |
|---|---|---|---|---|---|---|---|---|---|---|---|---|---|---|---|
| | α>.70 | CR>.70 | AVE>.50 | TRU | REC | REP | ABS | KSB | KSE | TRU | REC | REP | ABS | KSB | KSE |
| TRU | .89 | .92 | .75 | **.89** | | | | | | | | | | | |
| REC | .89 | .93 | .82 | .19 | **.90** | | | | | .84 | | | | | |
| REP | .90 | .93 | .77 | .72 | .18 | **.79** | | | | .79 | .82 | | | | |
| ABS | .92 | .94 | .81 | .72 | .26 | .69 | **.91** | | | .67 | .72 | .65 | | | |
| KSE | .93 | .95 | .82 | .57 | .23 | .61 | .58 | | **.88** | .66 | .77 | .67 | .63 | | |
| KSB | .78 | .86 | .61 | .66 | .34 | .70 | .29 | .66 | **.90** | .71 | .62 | .71 | .52 | .68 | |

*Note*: TRU: Trust, REC: Reciprocity, REP: Reputation, ABS: Ability to share, KSE: Knowledge Self-Efficacy, KSB: Knowledge Sharing Behaviour. α = Cronbach's alpha, CR = Composite reliability, AVE = Average variance extracted.

* Fornell-Larcker criterion: the value in **bold** is accepted if it is higher than the corresponding row and column values.

** HTMT Ratio < .85 is valid.

## 4.3 Construct cross-validated redundancy ($Q^2$)

The blindfolding output of SmartPLS is calculated to measure the predictive relevance of the latent variables of a study. Table 4 shows that Stone-Geisser $Q^2$ equal 1 –SSE/SSO. As a result of Henseler et al. [64] procedures, a research model with $Q^2 > 0$ attained the accepted value of predictive relevance.

**Table 3. Cross loading of constructs.**

| | ABS | KSB | KSE | REC | REP | TRU |
|---|---|---|---|---|---|---|
| ABS1 | **0.910** | 0.655 | 0.149 | 0.630 | 0.565 | 0.551 |
| ABS2 | **0.911** | 0.643 | 0.129 | 0.590 | 0.488 | 0.543 |
| ABS3 | **0.915** | 0.619 | 0.191 | 0.590 | 0.479 | 0.493 |
| ABS4 | **0.854** | 0.654 | 0.195 | 0.754 | 0.519 | 0.571 |
| KSB1 | 0.531 | **0.688** | 0.136 | 0.466 | 0.401 | 0.453 |
| KSB3 | 0.575 | **0.785** | 0.161 | 0.585 | 0.504 | 0.508 |
| KSB4 | 0.547 | **0.806** | 0.083 | 0.514 | 0.48 | 0.532 |
| KSB5 | 0.584 | **0.836** | 0.18 | 0.583 | 0.513 | 0.518 |
| KSE1 | 0.155 | 0.131 | **0.882** | 0.237 | 0.218 | 0.109 |
| KSE2 | 0.202 | 0.184 | **0.897** | 0.258 | 0.238 | 0.149 |
| KSE4 | 0.120 | 0.146 | **0.906** | 0.179 | 0.186 | 0.058 |
| KSE5 | 0.180 | 0.179 | **0.929** | 0.257 | 0.210 | 0.122 |
| REC1 | 0.673 | 0.623 | 0.256 | **0.906** | 0.560 | 0.570 |
| REC2 | 0.627 | 0.640 | 0.240 | **0.908** | 0.513 | 0.567 |
| REC3 | 0.649 | 0.617 | 0.213 | **0.909** | 0.509 | 0.554 |
| REP1 | 0.453 | 0.468 | 0.179 | 0.46 | **0.842** | 0.471 |
| REP2 | 0.559 | 0.575 | 0.245 | 0.536 | **0.908** | 0.520 |
| REP3 | 0.508 | 0.539 | 0.185 | 0.537 | **0.887** | 0.518 |
| REP4 | 0.483 | 0.553 | 0.216 | 0.500 | **0.873** | 0.480 |
| TRU1 | 0.552 | 0.553 | 0.123 | 0.496 | 0.498 | **0.850** |
| TRU2 | 0.537 | 0.584 | 0.130 | 0.554 | 0.462 | **0.868** |
| TRU3 | 0.487 | 0.549 | 0.075 | 0.572 | 0.505 | **0.883** |
| TRU4 | 0.509 | 0.547 | 0.101 | 0.528 | 0.502 | **0.865** |

**Table 4. Construct cross-validated redundancy.**

| Latent variable | SSO | SSE | $Q^2 = 1- (SSE/SSO)$ |
|---|---|---|---|
| Knowledge Sharing Behaviour | 1,632.00 | 1,112.16 | 0.318 |

*Note*: SSE is the sum of Squares of Prediction Errors; SSO is the Sum of Squares Observations.

## 4.4 Coefficient of determination ($R^2$)

The output of the PLS3 structure produced a coefficient of determination values ($R^2$) of knowledge sharing behaviour (KSB) as 0.695. This means that ABS, KSE, REC, REP, and TRU together explained 69.5% of knowledge-sharing behaviour among head nurses in Jordan. A larger $R^2$ value increases the predictive ability of the structural model. In the current study, $R^2$ is substantial according to Chin's [65] classification of $R^2$ value.

## 4.5 Study hypotheses testing

This study investigated four hypotheses concerning the moderation effect of knowledge self-efficacy between trust, reputation, reciprocity, and ability to share with knowledge-sharing behaviour. The result of 5000 bootstrapping of 283 cases to measure the significance of the path coefficients with a 95% Confidence Interval showed a moderation effect of knowledge self-efficacy in the relationship between trust, reputation, reciprocity, and knowledge-sharing behaviour, as shown in Table 5 and Fig 2.

In more detail, the moderating effect of knowledge self-efficacy (interaction between knowledge self-efficacy and trust, TRUST*KSE) exists in the relationship between reputation and knowledge-sharing behaviour. The results were also statistically significant ($\beta = 0.142$, $p = 0.03$) and positive, which revealed that knowledge self-efficacy was able to moderate the relationship between trust and knowledge-sharing behaviour positively. Based on these findings, trust was more positively effective on knowledge sharing behaviour when the knowledge self-efficacy is at a higher level; when the knowledge self-efficacy increases, this factor will increase; hence, trust will increase the knowledge sharing behaviour.

Concerning reciprocity, the results were statistically significant ($\beta = -0.167$, $p = 0.018$). It was also negative, meaning that knowledge self-efficacy negatively moderated the relationship between reciprocity and knowledge-sharing behaviour. This finding indicated that, at a high level of knowledge self-efficacy, reciprocity had a lower effect on knowledge-sharing behaviour and vice versa. In more detail, when the level of knowledge self-efficacy reduces, reciprocity would be more effective in knowledge-sharing behaviour.

**Table 5. Test of the moderating effect of knowledge self-efficacy.**

| # | Path | $f^2$ | $\beta$ | SE | T Value | P (Sig) |
|---|---|---|---|---|---|---|
| H1 | TRU*KSE---> KSB | 0.026 | 0.149 | 0.069 | 2.166 | 0.03 ** |
| H2 | REC*KSE---> KSB | 0.032 | -0.167 | 0.071 | 2.358 | 0.018 ** |
| H3 | REP*KSE---> KSB | 0.078 | 0.192 | 0.053 | 3.640 | 0.001 *** |
| H4 | ABS*KSE---> KSB | 0.015 | -0.119 | 0.066 | 1.794 | 0.073 [n.s] |

Note

***: p<0.01

**: p<0.05

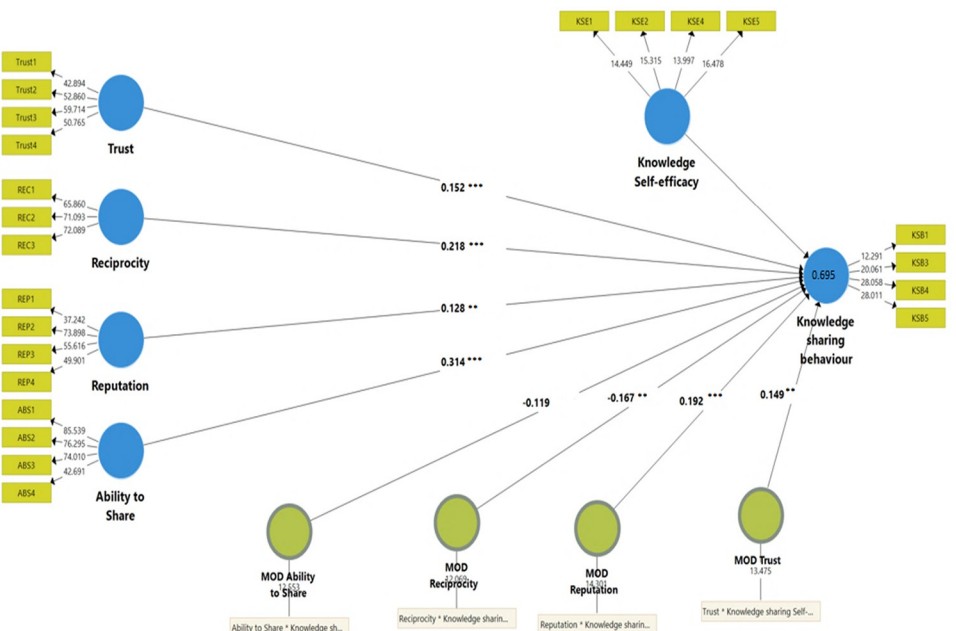

**Fig 2. SEM figure of knowledge sharing behaviour with moderating effect.**

Reputation was also found to be a moderator and statistically significant and positive ($\beta = 0.192$, $p < 0.001$). This revealed that knowledge self-efficacy positively moderated the relationship between reputation and knowledge-sharing behaviour. Thus, it can be concluded that reputation was more positively effective on knowledge-sharing behaviour at a high level of knowledge self-efficacy. Likewise, if knowledge self-efficacy increases, the reputation factor will affect the level of knowledge-sharing behaviour.

Surprisingly, the bootstrapping calculation between the ability to share and sharing behaviour did not have a significant effect ($\beta = -0.119$, $p = 0.073$). This means that knowledge self-efficacy did not moderate the relationship between the ability to share and knowledge-sharing behaviour.

## 5. Discussion

This is the first study investigating the knowledge-sharing behaviour of OHC in the Jordanian context. The main objective of this study was to assess the moderating effect of knowledge self-efficacy on the relationship between four individual factors and knowledge-sharing behaviour among head nurses in online health communities in Jordan.

Lai and Hsieh [55] found that reciprocity was a critical motivator of continued knowledge-sharing behaviour for people with low knowledge self-efficacy. First, they found that knowledge self-efficacy moderates trust and knowledge-sharing behaviour in online health communities. If an individual has a strong sense of knowledge self-efficacy, he or she will have no problem sharing [55].

The current study found that knowledge self-efficacy among head nurses can increase the effect of trust on knowledge-sharing behaviour and higher knowledge self-efficacy. The effect of trust on the part of head nurses is more positive and effective regarding their knowledge-sharing behaviour. In line with Social Cognitive Theory, this finding suggests that nursing knowledge-sharing behaviour increases with their ability to control or behave.

Second, knowledge self-efficacy served as a moderator between reciprocity and knowledge-sharing behaviour; this finding was consistent with previous studies in different contexts [37]. More specifically, the moderating effect of knowledge self-efficacy between reciprocity and knowledge-sharing behaviour implies that an individual with low knowledge self-efficacy is more reciprocal in sharing knowledge than an individual with a high score of knowledge self-efficacy.

Third, the present study found that knowledge self-efficacy moderates reputation and knowledge-sharing behaviour. This implies that the effect of reputation on knowledge-sharing behaviour was high for the employee with a high level of self-efficacy. In other words, reputation strongly influenced knowledge contributors with high levels of self-knowledge efficacy [3, 55]. This significance of the moderating role of knowledge sharing between reputation and knowledge sharing is also in line with social cognitive theory. As stated, the theory asserts that behaviour is the product of an individual's past experience and level of self-efficacy. Accordingly, knowledge self-efficacy increases the effectiveness of reputation in enhancing knowledge-sharing behaviours among head nurses. Head nurses who gain reputations from online communities and have higher knowledge self-efficacy will be more likely to share knowledge in OHCs. The present study extends the understanding of the moderating role of knowledge self-efficacy between reputation and knowledge-sharing behaviour. It also extends the understanding of the applicability of knowledge self-efficacy among head nurses working in online health communities, specifically in Jordan.

The results contradicted the proposed hypothesis, as knowledge self-efficacy did not mediate between the ability to share and knowledge-sharing behaviour. This result might be due to inadequate knowledge-sharing activities at private hospitals, which may have shown that knowledge self-efficacy does not support their ability to share in OHCs. In addition, this result is consistent with Sitharthan et al. [66] and Nguyen et al. [3] studies that reported that self-efficacy does not always moderate the relationship between two personal variables.

## 5.1 Implications and future research

This study expanded the literature regarding knowledge-sharing behaviour, individual factors, and knowledge self-efficacy. Examining the study model in the healthcare sector in Jordan is not only considered to offer an extension of the literature. However, it also fills the gap in the existing literature by providing a comprehensive understanding of the above moderating effect of knowledge self-efficacy, which could enrich knowledge-sharing behaviours.

The study's findings could benefit policymakers in hospital settings to improve the knowledge-sharing behaviour of head nurses in OHCs and help them understand essential factors that could affect their knowledge-sharing behaviour in these communities. Moreover, it is helpful for the Jordanian context in obtaining a better understanding of the main factors increasing the knowledge-sharing behaviour of head nurses. The role of individual factors includes trust, reciprocity, and reputation, which have been shown to improve knowledge sharing in online health communities.

Furthermore, it provides practical contributions about the role of knowledge self-efficacy as a predictor of knowledge-sharing behaviour. Specifically, the results regarding the link between knowledge self-efficacy and knowledge-sharing behaviour offer clear insights for hospital management to avoid challenges affecting knowledge-sharing practices [29, 67].

The health policymakers can apply these findings in setting a plan for supporting knowledge self-efficacy. For example, head nurses should allocate specific time to share knowledge via online healthcare communities and connect sharing amounts to a "points system." The knowledge-sharing behaviour among nurses depends on their willingness to acquire skills,

knowledge, and experiences through online communities. Other contextual factors, such as hospital size, also affect the nurses' knowledge; for instance, a large hospital would conduct effective training sessions, workshops, and seminars for nurses, while a small hospital would not. The management should encourage the knowledge-sharing behaviour culture through OHCs to increase the level of knowledge self-efficacy among head nurses. Topics in different disciplines, such as pain management, safety performance, quality care, and health digitalisation, should be considered in this regard. Accordingly, increasing knowledge-sharing behaviour among its communities. Overall, this study gives top management at private hospitals more understanding of how knowledge self-efficacy can encourage head nurses to share their knowledge in online health communities. Future research can explore new variables as independent, dependent, or moderating variables such as organisational and environmental factors [68, 69] or extend the investigation to more regions and sectors such as education and finance. Moreover, future research may investigate non-significant results in this study, such as the ability to share through the moderating effect of knowledge self-efficacy.

### 5.2 Study limitations

This study has limitations. First, the generalizability of the current study's findings is limited in two aspects. In particular, the study involved one representative from among the head nurses of each department in the hospitals. However, other employees were not considered when making up the study sample. Second, the data collection was restricted to private hospitals in Amman city due to the ability to access data. Therefore, the findings may not be generalisable to other sectors in Jordan or other countries. This could extend to other hospitals in different regions or other healthcare sectors in the future. Hence, comparable studies could be conducted in other sectors to consider more employees during the survey. Third, the study was completed in 2019; the data reported here were dated. However, because the study variables are interpersonal interactions, they are less likely to be affected by time [68]. The last limitation of the study is that it is cross-sectional and cannot establish the causality of the study model.

## 6. Conclusion

As technology and social media become advanced, knowledge-sharing behaviour is in OHCs to enhance the health status of individuals and communities. This study focused on knowledge self-efficacy and reflection on individuals' factors and knowledge-sharing behaviour. Head nurses with a high self-efficacy of knowledge can improve their knowledge-sharing behaviour. Knowledge self-efficacy moderates trust, reciprocity, and reputation with knowledge-sharing behaviour, while the ability to share did not. This study has several implications for private hospitals in Amman regarding the key roles of individual factors and knowledge self-efficacy in improving knowledge-sharing behaviour among head nurses in online health communities. Accordingly, recognising the links explained by the study model could add value to the theory and practice.

## Supporting information

**S1 Appendix.**
(DOCX)

**S1 Data.**
(XLS)

## Author Contributions

**Conceptualization:** Salah Shehab.

**Data curation:** Salah Shehab, Mohammed Dauwed, Badr K. Aldhmadi, Khalid Al-Mugheed.

**Formal analysis:** Salah Shehab, Mohammad Al-Bsheish, Mu'taman Jarrar.

**Investigation:** Ahmed Meri, Mohammed Dauwed, Badr K. Aldhmadi, Adi Alsyouf.

**Methodology:** Salah Shehab, Mohammad Al-Bsheish, Khalid Al-Mugheed, Mu'taman Jarrar.

**Project administration:** Mohammad Al-Bsheish.

**Resources:** Ahmed Meri, Badr K. Aldhmadi, Haitham Mohsin Kareem.

**Software:** Ahmed Meri, Mohammed Dauwed.

**Supervision:** Mohammad Al-Bsheish, Mu'taman Jarrar.

**Validation:** Ahmed Meri, Mohammed Dauwed, Badr K. Aldhmadi, Haitham Mohsin Kareem, Adi Alsyouf.

**Visualization:** Mohammed Dauwed, Badr K. Aldhmadi, Haitham Mohsin Kareem, Adi Alsyouf.

**Writing – original draft:** Salah Shehab, Mohammad Al-Bsheish.

**Writing – review & editing:** Mohammad Al-Bsheish, Badr K. Aldhmadi, Haitham Mohsin Kareem, Adi Alsyouf, Khalid Al-Mugheed, Mu'taman Jarrar.

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
