## [Decision Letter · Decision Letter 0]

20 Sep 2022

PONE-D-22-23050Knowledge Sharing Behaviour among Head Nurses in Online Health Communities: The Moderating Role of Knowledge Self-EfficacyPLOS ONE

Dear Dr. Jarrar,

Thank you for submitting your manuscript to PLOS ONE. After careful consideration, we feel that it has merit but does not fully meet PLOS ONE’s publication criteria as it currently stands. Therefore, we invite you to submit a revised version of the manuscript that addresses the points raised during the review process.

We look forward to receiving your revised manuscript.

Kind regards,

Supat Chupradit, Ph.D., M.Ed., B.Sc.(OT), B.P.A., B.Ed., B.A.

Academic Editor

PLOS ONE

Journal Requirements:

Reviewers' comments:

Reviewer's Responses to Questions

**Comments to the Author**

1. Is the manuscript technically sound, and do the data support the conclusions?

Reviewer #1: Yes

Reviewer #2: Yes

Reviewer #3: Yes

Reviewer #4: Yes

Reviewer #5: Yes

2. Has the statistical analysis been performed appropriately and rigorously? 

Reviewer #1: Yes

Reviewer #2: Yes

Reviewer #3: Yes

Reviewer #4: Yes

Reviewer #5: Yes

3. Have the authors made all data underlying the findings in their manuscript fully available?

Reviewer #1: Yes

Reviewer #2: Yes

Reviewer #3: Yes

Reviewer #4: Yes

Reviewer #5: Yes

4. Is the manuscript presented in an intelligible fashion and written in standard English?

Reviewer #1: Yes

Reviewer #2: Yes

Reviewer #3: Yes

Reviewer #4: Yes

Reviewer #5: Yes

5. Review Comments to the Author

Reviewer #1: 1-Abstract part: the purpose fo this study is not related with study hypothesis. The purpose of this study might be examine or investigate associated between knowledge self efficacy and individual factors.

2-According to sampling design that is calculated sampling sizes by using G*Power the authors should describe the effect size, power of test.

3-Very well organize in result and discussion part.

Reviewer #2: Overall, this manuscript is quite well written. However, it is also found that some references, especially in literature reviews, are quite old. Authors should focus only on research in the past 5 years and consider citations to these relevant articles.

 Sriyakul, T., & Jermsittiparsert, K. (2021). Factors effecting Preventive Health Behavior among the Students at Universities in Thailand: Mediating Role of Self Efficacy. Educational Sciences: Theory & Practice, 21(4), 223-233.

 Rodboonsong, S., & Sawasdee, A. (2020). Fostering Knowledge Sharing Behavior in Educational Institutes of Thailand. International Journal of Crime, Law and Social Issues, 7(2), 63-73.

 Jarinto, K., Jermsittiparsert, K., & Chienwattanasook, K. (2019). A Theoretical and Empirical Framework for Knowledge Sharing: An Auto Industry Case-study. International Journal of Innovation, Creativity and Change, 10(1), 406-425.

Reviewer #3: I have reviewed the paper title: Knowledge Sharing Behaviour among Head Nurses in Online Health Communities: The Moderating Role of Knowledge Self-Efficacy

This article is strongly recommended for publication after incorporating certain changes. This article needs thorough proofreading. The overall quality of the Language is good. Just major grammatical mistakes are found. All tables and figures are relevant. The research Methodology has been well defined. All data are aligned with the findings of the research. This article is a good attempt in field research and will be beneficial for future researchers.

Abstract

1. Understand, Clear

Introduction

1. Lack of Gap reflection to do clear research, please clarify the point. Why point to internet contexts?

Methods

1. Measuring of research by your research approaches that methods are there any references to reduce bias? Please explain, Is it a limitation in interpreting the results to populations/samples?

2. Should describe the population, random sampling and the sample size to be concise and clearer and add academic support description.

3. Do you have IRB approval in this research?, If it have please show number approve, If it not please explain how about your method to protect participants in this research.

4. Please add Data analysis section: The key statistics that they be used in hypothesis testing, should be described in detail to support your decision at the end of this section.

5. Recheck table / figure to quality standard for the journal.

6. Suggestion and Policy recommendation, please add and point it in your paper.

7. Limitation of your study?, Add recommendations about policy recommend.

8. References, the researcher should check and revise the format again. Check styles and recheck all.

Reviewer #4: Overall

The researcher writes and organizes the article content well, clearly and appropriately according to the academic context.

Literature review

Figure should be adjusted to make it more noticeable and clearly according to publication standards. please check resolution and figure standard again.

Method

Do you have IRB approval in this research?, If it have please show number approve, If it not please explain how about your method to protect participants in this research.

Result

4.2 Model Assessment

1) The researcher should state the statistical criteria which used to accept the outer and inner model.

2) The result should be show the SEM figure with important statistical values in the model: outer and inner model.

Reference

The researcher should check and revise the format again.

Reviewer #5: -Why do you choose head nurses?

-The random sampling should be clearly stated: which random sampling method is applied?

-Should be specified Validity and Reliability of questionnaire.

-Knowledge-sharing behaviour may be developed from skills, knowledge, and experiences of the nurses themselves. The more knowledge the nurses have, the more their knowledge-sharing behavior develop. In addition, the size of the hospital also affects the knowledge of the nurses, for example, a large hospital would conduct effective training sessions for nurses, while a small hospital would not.

-The research findings should be compared with other areas in Jordan.

6. PLOS authors have the option to publish the peer review history of their article (what does this mean?). If published, this will include your full peer review and any attached files.

Reviewer #1: No

Reviewer #2: **Yes: **Kittisak JERMSITTIPARSERT

Reviewer #3: No

Reviewer #4: No

Reviewer #5: No

---

## [Author Response · Author response to Decision Letter 0]

9 Oct 2022

The following is our point-by-point response.

Reviewer #1: 

Here are the responses to your concerns:

1. Abstract part: the purpose of this study is not related with study hypothesis. The purpose of this study might be examine or investigate associated between knowledge self-efficacy and individual factors.

Thank you for the comments. The study purpose adjusted accordingly. 

2. According to sampling design that is calculated sampling sizes by using G*Power the authors should describe the effect size, power of test.

Thank you for the comment. Edited accordingly. 

3. Very well organize in result and discussion part.

Thank you. 

Reviewer #2:

1. Overall, this manuscript is quite well written. However, it is also found that some references, especially in literature reviews, are quite old. Authors should focus only on research in the past 5 years and consider citations to these relevant articles..

Thank you for the comment. Corrected accordingly and the following references added. 

-Rodboonsong, S., & Sawasdee, A. (2020). Fostering Knowledge Sharing Behavior in Educational Institutes of Thailand. International Journal of Crime, Law and Social Issues, 7(2), 63-73.

Jarinto, K., Jermsittiparsert, K., & Chienwattanasook, K. (2019). A Theoretical and Empirical Framework for Knowledge Sharing: An Auto Industry Case-study. International Journal of Innovation, Creativity and Change, 10(1), 406-425.

Reviewer #3:

1. This article is strongly recommended for publication after incorporating certain changes. This article needs thorough proofreading. The overall quality of the Language is good. Just major grammatical mistakes are found. All tables and figures are relevant. The research Methodology has been well defined. All data are aligned with the findings of the research. This article is a good attempt in field research and will be beneficial for future researchers.

Thank you for the comments provided and efforts in optimizing our manuscript. The language corrected accordingly and sent again to native English speaker.

2. Abstract: Understand, Clear.

Thank you.

3. Introduction: Lack of Gap reflection to do clear research, please clarify the point. Why point to internet contexts?

Thank you for the comment. Corrected accordingly and become more reader friendly. Online Health Communities (OHCs) are one kind of an Online Communities, where maintaining health information is a public concern. OHCs through social media and other web-based forums, facilitate their members to participate in health topics, even those with sensitive considerations such as pregnancy, menstruation, and sexuality (Fan et al., 2014; Rai et al., 2012). Several studies have identified the role of individual factors in knowledge-sharing behaviours (Abdel Fattah et al., 2020; Fullwood et al., 2019; Obrenovic et al., 2020). However, lack of studies exploring the role of knowledge self-efficacy between the associations of individual factors (trust, reciprocity, reputation, and ability to share) with knowledge-sharing behaviours.

4. Measuring of research by your research approaches that methods are there any references to reduce bias? Please explain, Is it a limitation in interpreting the results to populations/samples?

Thank you for the comment. Corrected accordingly and study limitations improved.

5. Should describe the population, random sampling and the sample size to be concise and clearer and add academic support description.

Thank you for the comment. Corrected accordingly and become more reader friendly. The population of this study was private hospitals’ head nurses in Amman, Jordan. Private hospitals in the capital (i.e. Amman) because these institutions have competitive advantages, technological capabilities, highest capacity, diversity in terms of specialty, supportive research cultures, and the highest number of hospitals located in Amman (n= 32). The research team attained approval from 22 private hospitals with 322 head nurses (study population). The sample size was calculated based on the G*Power software package, effect size (f 2= 0.15); a significant level (α= 0.05) and power 1-β = 0.95, which calculated that a minimum sample size of 74 was required with five independent variables, including the moderator.

6. Do you have IRB approval in this research? If it have please show number approve, If it not please explain how about your method to protect participants in this research..

Thank you for the comment. Corrected accordingly and become more reader friendly. Ethical approval number UNITEN/COGS 23/2/1/PM20604 was attained from the College of Graduate Studies, Universiti Tenaga Nasional, Malaysia, on 25 April 2018. The hospitals approved their employees' voluntary participation in the study and encouraged their head nurses to participate, and informed consent was obtained from all head nurses agreed to be part of this survey.

7. Please add Data analysis section: The key statistics that they be used in hypothesis testing, should be described in detail to support your decision at the end of this section.

Thank you for the comment. Corrected accordingly and become more reader friendly. Descriptive statistics and moderation regression analysis using SPSS and structural equation modelling approach (i.e. Smart PLS-SEM, Version 3) were the key statistics in this study; respectively. This study used Smart PLS3 to test the hypotheses posited. Smart PLS3 uses a bootstrapping technique to estimate path coefficients and standard errors (Awang et al., 2015). The moderation impact of self-efficacy was evaluated using a 5000 bootstrap sample, a 95% confidence interval (CI) and significance level of 0.05. Before running Smart PLS3, descriptive results were performed using SPSS Version 18.0 (SPSS Inc., Chicago, IL, USA).

8. Recheck table / figure to quality standard for the journal.

Thank you for the comment. The table / figure has been redrawn to make it clearer

9. Suggestion and Policy recommendation, please add and point it in your paper.

Policy recommendations has improved and revised accordingly

10. Limitation of your study? Add recommendations about policy recommend.

Limitation and recommendations has improved and revised accordingly

11. References, the researcher should check and revise the format again. Check styles and recheck all.

Thank you for the comment. References and study format has revised accordingly.

Reviewer #4:

The researcher writes and organizes the article content well, clearly and appropriately according to the academic context.

1. Literature review: Figure should be adjusted to make it more noticeable and clearly according to publication standards. Please check resolution and figure standard again.

Thank you for the comment. The model (Figure 1) has been redrawn to make it clearer

2. Method: Do you have IRB approval in this research?, If it have please show number approve, If it not please explain how about your method to protect participants in this research.

Thank you for the comment. Corrected accordingly and become more reader friendly. Ethical approval number UNITEN/COGS 23/2/1/PM20604 was attained from the College of Graduate Studies, Universiti Tenaga Nasional, Malaysia, on 25 April 2018. The hospitals approved their employees' voluntary participation in the study and encouraged their head nurses to participate, and informed consent was obtained from all head nurses agreed to be part of this survey.

3. Result: 4.2 Model Assessment

a) The researcher should state the statistical criteria which used to accept the outer and inner model.

Thank you for the comment. Corrected accordingly and become more reader friendly. Descriptive statistics and moderation regression analysis using SPSS and structural equation modelling approach (i.e. Smart PLS-SEM, Version 3) were the key statistics in this study; respectively. This study used Smart PLS3 to test the hypotheses posited. Smart PLS3 uses a bootstrapping technique to estimate path coefficients and standard errors (Awang et al., 2015). The moderation impact of self-efficacy was evaluated using a 5000 bootstrap sample, a 95% confidence interval (CI) and significance level of 0.05. Before running Smart PLS3, descriptive results were performed using SPSS Version 18.0 (SPSS Inc., Chicago, IL, USA).

b) The result should be show the SEM figure with important statistical values in the model: outer and inner model.

Thank you for the comment. SEM figure has added as Figure 2 shown 

4. Reference: The researcher should check and revise the format again.

Thank you for the comment. References and study format has revised accordingly.

Reviewer #5: 

1. Why do you choose head nurses?

Thank you for the comment. Justification has added in methodology part accordingly. 

“Head nurses were targeted in this study to serve the study purposes as they considered as one of health leaders. Public and even followers’ nurses prefer contact with head nurses due to their managerial rank as a first line management to them; thus, they are knowledgeable, closer, and often participating part in online communities”.

2. The random sampling should be clearly stated: which random sampling method is applied?

3. Should be specified Validity and Reliability of questionnaire.

Thank you for the comments. Reliability and validity reported accordingly (α= Cronbach's alpha, CR = Composite reliability)

4. Knowledge-sharing behaviour may be developed from skills, knowledge, and experiences of the nurses themselves. The more knowledge the nurses have, the more their knowledge-sharing behavior develop. In addition, the size of the hospital also affects the knowledge of the nurses, for example, a large hospital would conduct effective training sessions for nurses, while a small hospital would not.

Nice paragraph and insight the authors to consider it in the discussion part. Thank you 

5. The research findings should be compared with other areas in Jordan.

Thanks your comments, authors did some text modifications. However, scarce research is available in the Jordanian context in this regard

We confirm that this work is non-funded and original and has not been published anywhere, nor is it currently under consideration for publication elsewhere.

Please address all correspondence concerning this manuscript to me at [mutaman.jarrar@yahoo.com, mkjarrar@iau.edu.sa].

Thank you for your consideration of this manuscript. 

Sincerely,

Mu’taman Jarrar

---

## [Decision Letter · Decision Letter 1]

31 Oct 2022

PONE-D-22-23050R1Knowledge Sharing Behaviour among Head Nurses in Online Health Communities: The Moderating Role of Knowledge Self-EfficacyPLOS ONE

Dear Dr. Jarrar,

Thank you for submitting your manuscript to PLOS ONE. After careful consideration, we feel that it has merit but does not fully meet PLOS ONE’s publication criteria as it currently stands. Therefore, we invite you to submit a revised version of the manuscript that addresses the points raised during the review process

We look forward to receiving your revised manuscript.

Kind regards,

Supat Chupradit, Ph.D., M.Ed., B.Sc.(OT), B.P.A., B.Ed., B.A.

Academic Editor

PLOS ONE

Journal Requirements:

Reviewers' comments:

Reviewer's Responses to Questions

**Comments to the Author**

1. If the authors have adequately addressed your comments raised in a previous round of review and you feel that this manuscript is now acceptable for publication, you may indicate that here to bypass the “Comments to the Author” section, enter your conflict of interest statement in the “Confidential to Editor” section, and submit your "Accept" recommendation.

Reviewer #1: All comments have been addressed

Reviewer #2: All comments have been addressed

Reviewer #3: All comments have been addressed

Reviewer #4: All comments have been addressed

2. Is the manuscript technically sound, and do the data support the conclusions?

Reviewer #1: Yes

Reviewer #2: Yes

Reviewer #3: Yes

Reviewer #4: Yes

3. Has the statistical analysis been performed appropriately and rigorously? 

Reviewer #1: No

Reviewer #2: Yes

Reviewer #3: Yes

Reviewer #4: Yes

4. Have the authors made all data underlying the findings in their manuscript fully available?

Reviewer #1: Yes

Reviewer #2: Yes

Reviewer #3: Yes

Reviewer #4: Yes

5. Is the manuscript presented in an intelligible fashion and written in standard English?

Reviewer #1: Yes

Reviewer #2: Yes

Reviewer #3: Yes

Reviewer #4: Yes

6. Review Comments to the Author

Reviewer #1: 1-The social cognitive theory that is found since 1996, that might be questionable this theory still suitable for using at the present. The authors might illustrulate the evidences support.

2-It not nesssary show the hyphothesis setting process of this study. That might be shows in review literature part.

3-Due to using the part analysis. This statistic is the family of multivariate statistic. The authors should choose another method for calculate minimum sampling size.

4-There are some measure are established since 1999 the authors should examine the psychometric properties before using collect data.

5-In this study has found the lower Beta values (see in table 5), the authors should explain the reasons and evidences support why the knowledge self-efficacy is not mediate between the ablity.

Reviewer #2: The authors have made complete revisions to the paper to the satisfaction of all recommendations. The main strength of this paper is its strong literary references. However, the authors may also modify the Abstract composition to be a single paragraph without any subheading in this section.

Reviewer #3: The manuscript revision and response by author, Knowledge Sharing Behaviour among Head Nurses in Online Health Communities: The Moderating Role of Knowledge Self-Efficacy. I think you improve all comments that I reviewed this manuscript. Please recheck all references.

Regards,

Reviewer #4: Knowledge Sharing Behaviour among Head Nurses in Online Health Communities: The Moderating Role of Knowledge Self-Efficacy. Thank you for your work hard to revise manuscript. I follow your response referees. I think Its improve to valuable article to publish.

Best Regards,

7. PLOS authors have the option to publish the peer review history of their article (what does this mean?). If published, this will include your full peer review and any attached files.

Reviewer #1: No

Reviewer #2: **Yes: **Kittisak JERMSITTIPARSERT

Reviewer #3: No

Reviewer #4: No

---

## [Author Response · Author response to Decision Letter 1]

3 Nov 2022

Journal Requirements:

Thank you for the comment. 

References and study format has revised accordingly, and all unrelated references were removed accordingly.

Reviewer #1: 

1- The social cognitive theory that is found since 1996, that might be questionable this theory still suitable for using at the present. The authors might illustrate the evidences support.

Thank you for the comment.

The social cognitive theory (SCT) is frequently used to guide behavior change interventions such as Knowledge Sharing Behaviour. It may be particularly useful for examining how individuals interact with their surroundings. The SCT can be used to understand the influence of social determinants of health and a person's experiences on behavior change. Comparing to other theories in this regard such as social exchange theory was developed in 1958, SCT is still current and many recent articles were used this theory.

for example:

Lin, H. C., & Chang, C. M. (2018). What motivates health information exchange in social media? The roles of the social cognitive theory and perceived interactivity. Information & Management, 55(6), 771-780.‏

Ahmed, Y. A., Ahmad, M. N., Ahmad, N., & Zakaria, N. H. (2019). Social media for knowledge-sharing: A systematic literature review. Telematics and informatics, 37, 72-112.‏

Lin, X., & Kishore, R. (2021). Social media-enabled healthcare: a conceptual model of social media affordances, online social support, and health behaviors and outcomes. Technological Forecasting and Social Change, 166, 120574.‏

2- It not necessary shows the hypotheses setting process of this study. That might be shows in review literature part.

Thank you. Hypotheses testing and study framework was established in the literature review part 

3- Due to using the part analysis. This statistic is the family of multivariate statistic. The authors should choose another method for calculate minimum sampling size.

Thank you for this comment. 

It was calculated by using another method and added to the manuscript accordingly. “Beside G*Power software package, Krejcie and Morgan, (1970) sample size formula was used to get minimum number of respondents to be surveyed, which is 210”.

4- There are some measure are established since 1999 the authors should examine the psychometric properties before using collect data.

The reviewer means “Jarvenpaa, S. L., & Leidner, D. E. (1999). Communication and trust in global virtual teams. Organization science, 10(6), 791-815”

Despite this paper being old to some extent, however, it strong and citable one (more than 4000 citations). Moreover, the psychometric properties of this measure such as validity and reliability were examined in this study and it was valid and reliable. 

5- In this study has found the lower Beta values (see in table 5), the authors should explain the reasons and evidences support why the knowledge self-efficacy is not mediate between the ability.

Yes, the study empirical results show no moderation effect of knowledge self-efficacy between the ability and knowledge-sharing behaviors. 

Justification in the text is “This result might be due to inadequate knowledge-sharing activities at private hospitals, which may have shown that knowledge self-efficacy does not support their ability to share in OHCs. In addition, this result is consistent with Sitharthan et al. (65) and Nguyen et al. (3) studies that reported that self-efficacy does not always moderate the relationship between two personal variables”

Reviewer #2: 

The authors have made complete revisions to the paper to the satisfaction of all recommendations. The main strength of this paper is its strong literary references. However, the authors may also modify the Abstract composition to be a single paragraph without any subheading in this section

Thank you for this comment. Abstract become a single paragraph without any subheading 

Reviewer #3: 

The manuscript revision and response by author, Knowledge Sharing Behaviour among Head Nurses in Online Health Communities: The Moderating Role of Knowledge Self-Efficacy. I think you improve all comments that I reviewed this manuscript. Please recheck all references.

Thank you for your reviewing our work. References and study format has revised accordingly, and all unrelated references were removed accordingly.

Reviewer #4: 

Knowledge Sharing Behaviour among Head Nurses in Online Health Communities: The Moderating Role of Knowledge Self-Efficacy. Thank you for your work hard to revise manuscript. I follow your response referees. I think Its improve to valuable article to publish.

Thank you for your reviewing our work 

We confirm that this work is non-funded and original and has not been published anywhere, nor is it currently under consideration for publication elsewhere.

Note: The data uploaded in the system to confirm the availability for future researchers requesting access to the data from corresponding authors. 

Please address all correspondence concerning this manuscript to me at [mutaman.jarrar@yahoo.com, mkjarrar@iau.edu.sa].

Thank you for your consideration of this manuscript. 

Sincerely,

Mu’taman Jarrar

---

## [Decision Letter · Decision Letter 2]

22 Nov 2022

Knowledge Sharing Behaviour among Head Nurses in Online Health Communities: The Moderating Role of Knowledge Self-Efficacy

PONE-D-22-23050R2

Dear Dr. Jarrar,

We’re pleased to inform you that your manuscript has been judged scientifically suitable for publication and will be formally accepted for publication once it meets all outstanding technical requirements.

Kind regards,

Supat Chupradit, Ph.D., M.Ed., B.Sc.(OT), B.P.A., B.Ed., B.A.

Academic Editor

PLOS ONE

Additional Editor Comments (optional):

Reviewers' comments:

Reviewer's Responses to Questions

**Comments to the Author**

1. If the authors have adequately addressed your comments raised in a previous round of review and you feel that this manuscript is now acceptable for publication, you may indicate that here to bypass the “Comments to the Author” section, enter your conflict of interest statement in the “Confidential to Editor” section, and submit your "Accept" recommendation.

Reviewer #1: All comments have been addressed

Reviewer #3: All comments have been addressed

Reviewer #4: All comments have been addressed

2. Is the manuscript technically sound, and do the data support the conclusions?

Reviewer #1: Yes

Reviewer #3: Yes

Reviewer #4: Yes

3. Has the statistical analysis been performed appropriately and rigorously? 

Reviewer #1: Yes

Reviewer #3: Yes

Reviewer #4: Yes

4. Have the authors made all data underlying the findings in their manuscript fully available?

Reviewer #1: Yes

Reviewer #3: Yes

Reviewer #4: Yes

5. Is the manuscript presented in an intelligible fashion and written in standard English?

Reviewer #1: Yes

Reviewer #3: Yes

Reviewer #4: Yes

6. Review Comments to the Author

Reviewer #1: The authors organized the revise article very well. The information from this article are rich and usuful.

Reviewer #3: Knowledge Sharing Behaviour among Head Nurses in Online Health Communities: The Moderating Role of Knowledge Self-Efficacy. Revision version base on comments by reviewers. Accept.

Reviewer #4: Thank you for considering reviewer comments and suggestions. I am satisfied with the responses.

All the best for your article.

7. PLOS authors have the option to publish the peer review history of their article (what does this mean?). If published, this will include your full peer review and any attached files.

Reviewer #1: No

Reviewer #3: No

Reviewer #4: No

---

## [Editor Report · Acceptance letter]

28 Nov 2022

PONE-D-22-23050R2 

Knowledge Sharing Behaviour among Head Nurses in Online Health Communities: The Moderating Role of Knowledge Self-Efficacy 

Dear Dr. Jarrar:

I'm pleased to inform you that your manuscript has been deemed suitable for publication in PLOS ONE. Congratulations! Your manuscript is now with our production department. 

Kind regards, 

on behalf of

Assistant Professor Supat Chupradit 

Academic Editor

PLOS ONE